# Supporting COVID-19 policy-making with a predictive epidemiological multi-model warning system

Martin Bicher [1,2], Martin Zuba[3], Lukas Rainer[3], Florian Bachner [3], Claire Rippinger [2], Herwig Ostermann[3,4], Nikolas Popper[1,2,5], Stefan Thurner [6,7,8] & Peter Klimek [6,7 ✉]

## Abstract

**Background** In response to the SARS-CoV-2 pandemic, the Austrian governmental crisis unit commissioned a forecast consortium with regularly projections of case numbers and demand for hospital beds. The goal was to assess how likely Austrian ICUs would become over-burdened with COVID-19 patients in the upcoming weeks.

**Methods** We consolidated the output of three epidemiological models (ranging from agent-based micro simulation to parsimonious compartmental models) and published weekly short-term forecasts for the number of confirmed cases as well as estimates and upper bounds for the required hospital beds.

**Results** We report on three key contributions by which our forecasting and reporting system has helped shaping Austria's policy to navigate the crisis, namely (i) when and where case numbers and bed occupancy are expected to peak during multiple waves, (ii) whether to ease or strengthen non-pharmaceutical intervention in response to changing incidences, and (iii) how to provide hospital managers guidance to plan health-care capacities.

**Conclusions** Complex mathematical epidemiological models play an important role in guiding governmental responses during pandemic crises, in particular when they are used as a monitoring system to detect epidemiological change points.

## Plain language summary

During the SARS-CoV-2 pandemic, health authorities make decisions on how and when to implement inter-ventions such as social distancing to avoid overburdening hospitals and other parts of the healthcare system. We combined three mathematical models developed to predict the expected number of confirmed SARS-CoV-2 cases and hospitalizations over the next two weeks. This pro-vides decision-makers and the gen-eral public with a combined forecast that is usually more accurate than any of the individual models. Our forecasting system has been used in Austria to decide when to strengthen or ease response measures.

[1] Institute of Information Systems Engineering, TU Wien, Favoritenstraße 8-11, A-1040 Vienna, Austria. [2] dwh simulation services, dwh GmbH, Neustiftgasse 57-59, A-1070 Vienna, Austria. [3] Austrian National Public Health Institute, Stubenring 6, A-1010 Vienna, Austria. [4] Private University for Health Sciences, Medical Informatics and Technology GmbH, UMIT, Eduard-Wallnöfer-Zentrum 1, A-6060 Hall in Tirol, Austria. [5] Association for Decision Support Policy and Planning, DEXHELPP, Neustiftgasse 57-59, A-1070 Vienna, Austria. [6] Section for Science of Complex Systems, Medical University of Vienna, Spitalgasse 23, A-1090 Vienna, Austria. [7] Complexity Science Hub Vienna, Josefstädterstraße 39, A-1080 Vienna, Austria. [8] Santa Fe Institute, 1399 Hyde Park road, Santa Fe, NM 87501, USA. ✉email: peter.klimek@meduniwien.ac.at

The first known COVID-19 cases in Austria appeared at the end of February 2020 together with one of the first European superspreading events in the Tyrolean tourist region of Ischgl, visited by travellers from all over the globe[1]. In the first half of March 2020, a nationwide spread of the virus occurred with an exponential rise of confirmed cases[2]. These developments occurred against the dramatic backdrop of the neighbouring country of Italy, where despite strict non-pharmaceutical interventions (NPIs) case numbers kept surging, hospital capacities were exceeded and the military had to assist the health authorities[3,4]. To understand how likely similar developments would have been in Austria, mid-March a forecast consortium was formed and tasked by the government with a weekly forecasting of the expected developments in case numbers and how these developments would translate into demand for healthcare resources. The overarching policy goal at this stage was to navigate the crisis without overburdening the Austrian healthcare system. Over the summer, this was also given a legal basis with a clause in the Austrian COVID law that stay-at-home orders may only be implemented if healthcare capacities are in danger of becoming exhausted[5]. The Austrian Corona Commission, an advisory committee to the minister of health tasked with assessment of epidemiological risk, defined that ICUs would be able to cope with situations in which up to 33% of all ICU beds would be occupied by COVID-19 patients[6].

Austria took a series of non-pharmaceutical interventions (NPIs) in response to the crisis[7] during the first wave. Next to a ramping up of healthcare and public health capacities, airport restrictions and landing bans intensified in the first week of March. Gatherings were limited to 500 persons, cultural and other events started to be cancelled on March 10. On March 16, Austria went into a full lockdown with schools, bars, restaurants, and shops being closed, as well as a transitioning into home office for all non-essential employees[7]. Together with other, earlier measures, these NPIs effectively led to a rapid reduction of daily infection numbers. The number of new cases per day reached a first peak on March 26 with 1,065 cases[8]. In the first wave, COVID-19 related hospitalizations peaked on March 31 with 912 regular beds, whereas the ICU utilization peaked on April 8 with 267 beds (roughly 10% of the overall capacity) being occupied by COVID-19 patients. Daily new cases decreased over April after which they fluctuated at values below one hundred until July[9].

Starting in July 2020, case numbers in Austria started to increase again leading to a second wave in October. In response to this rise of case numbers, the Austrian government implemented a series of lockdowns with varying levels of stringency since beginning of November 2020. Hospitalizations peaked in the end of November, with 3,985 regular beds occupied on November 24, and 709 ICU beds occupied on November 25, respectively. According to officially reported data, this peak brought Austrian hospitals very close to the critical limit of 33% ICU bed occupancy by COVID patients.

The Austrian COVID-19 forecast consortium provided weekly short-term forecasts for case numbers and required hospital beds. In particular, the role of the forecast consortium was to forecast how likely the 33% threshold of ICU beds being occupied by COVID-19 patients would be crossed within our forecast horizon. Our consortium consisted of three independent modelling teams with experience in the use and development of mathematical and computational models to address epidemiological and public health challenges[10–16]. The consortium was complemented with experts from the Ministry of Health, the Austrian Agency for Health and Food Safety, as well as external public health experts in weekly meetings.

A plethora of epidemiological models to forecast the spread of COVID-19 has been proposed recently[17–22]. In the forecast consortium, we consolidated the output of three models into a single forecast of case numbers for 8–14 days and used these case numbers to predict the numbers of required hospital and ICU beds for 21 days for the country as a whole and for each of its nine federal states. In addition to these point estimates, we also provided upper and lower bounds for these numbers at various levels of confidence. These upper bounds of the hospital bed forecasts served as a guidance system for the regional hospital managers, allowing them to estimate how many beds should be reserved for COVID-19 patients if they were willing to accept a given level of risk. These forecasts have been published each week on the homepage of the Ministry of Health[23].

The idea of using a harmonized epidemiological forecast became popular for influenza prognoses[24] and was recently also adapted for COVID-19 forecasts by the European Center for Disease Control[25]. For influenza, the strategy has already been shown to be superior to results of individual models with respect to forecasting errors. Moreover, since the results are highly relevant for policy-making, this strategy also allowed for risk-sharing between the involved institutions making up the consortium.

While the SARS-CoV-2 pandemic has triggered an explosive growth of epidemiological forecasting models, substantially less research has been performed regarding how the results of such models should be disseminated for decision support[26–28]. In this work, therefore, we present the forecast and reporting system we developed based on the three independent forecasting models to support policy making in Austria. While the individual models have been adapted from pre-existing works, our main novelty lies in developing a reporting system to communicate relevant output to non-technical experts and to inform decisions regarding strengthening or easing NPIs.

After a brief summary of the individual models and strategies to combine their output, we describe the accuracy of our forecasts and how this accuracy depended on the phase of the epidemics (i.e., in high or low incidence phases, during waves, etc.). We consistently find that the combined forecasts were more accurate than any of the individual forecasts for three different strategies of combining them. Susbstantial over- or underestimations of the actual development typically occurred in all models simultaneously and sometimes signalled epidemiological change points. We discuss how our results were received by policy-makers, stakeholders in the healthcare system, and the public. We outline the main contributions of our approach to chart a safe path to re-open the country after the first lockdown and how the system informed the necessity for a second lockdown in November 2020. The aim of this work is to communicate the methods applied and developed which allowed three individually thinking modelling and simulation research units to work together in a joint task force producing a consolidated forecast, the benefits and shortcomings of the process, and the political impact of the achieved results. We conclude that epidemiological models can be useful as the basis for short-term forecast-based monitoring systems to detect epidemiological change points which in turn inform on the necessity to strengthen or ease NPIs.

## Methods
We used three conceptually different epidemiological COVID-19 models, developed and operated individually by three research institutions, namely a modified SIR-X differential equation model (Medical University of Vienna / Complexity Science Hub), an Agent-Based simulation model (TU Wien/dwh GmbH), and an epidemiological state space model (Austrian National Public Health Institute).

**Data**. Although the three models use different parameters and parametrization routines, they are calibrated using the same data

to generate weekly forecasts. Consequently, differences between the model forecasts are a result of different model structure and calibration, but not a result of different data sources. The models also used different nowcasting approaches to correct for late reporting of positive test results. We used data from the official Austrian COVID-19 disease reporting system (EMS[9]). The system is operated by the Austrian Ministry of Health, the federal administrations, and the Austrian Agency for Health and Food Safety.

For every person tested positively in Austria, the EMS contains information on the date of the test, date of recovery or death, age, sex and place of residence. Furthermore, hospital occupancy of COVID-19 patients in ICU and normal wards are available from daily reports collected by the Ministry for Internal Affairs.

We declare that our research does not require ethics approval. We used an existing secondary dataset maintained as database to collect the information needed. It includes anonymous information and it is not possible to link data in order to generate identifiable information. Data are properly anonymised. The information is legally accessible and appropriately protected by law.

**Extended SIR-X model**. One of our models is an extension of the recently introduced SIR-X model[17]. The original SIR-X model introduced a parsimonious way to extend the classic SIR dynamics with the impact of NPIs. In particular, two classes of NPIs are considered. First, there are NPIs that lead to a contact reduction of all individuals (susceptible and infected ones). Such NPIs include social distancing and other lockdown measures. Second, the model also represents NPIs that reduce the effective duration of infectiousness for infected individuals. Contact-tracing and quarantine belong to this category.

The model was extended by adding mechanisms by which susceptible but quarantined individuals increase their number of contacts again as well as waning immunity; we refer to this model as the XSIR-X model. The model further includes an age-structured population. Model parameters are calibrated using a numerical optimization procedure that is separately performed in multiple time windows corresponding to phases with different regimes of NPIs. Whether changes in NPIs indeed led to substantial changes in behaviour was inferred from mobility data to identify such calibration time windows[29,30]. A more detailed technical model description can be found in Supplementary Note 2.

**Agent-based SEIR model**. The second model is an Agent-Based SEIR type model, furthermore abbreviated as AB model. In this section we will only give a rough overview of this model. For a detailed and technical model description (about 20 pages long) including all used parameter values we refer to previous published work (see supplemental material in [13]). Since the model is subject to continuous improvements, the model description is continuously updated and found at http://www.dwh.at/en/projects/covid-19/.

The AB model is stochastic, population-dynamic and depicts every inhabitant of Austria as one model agent. It uses sampling methods to generate an initial agent population with statistically representative demographic properties and makes use of a partially event-based, partially time-step (1 day)-based update strategy to enhance in time.

The model is based on a validated population model of Austria including demographic processes like death, birth, and migration[15]. Contacts between agents are responsible for disease transmission and are sampled via locations in which agents meet: schools, workplaces, households and leisure-time. After being

infected, agents go through a detailed disease and/or patient pathway that depicts the different states of the disease and the treatment of the patient.

The model input consists of a time-line of modelled NPIs; parameters are calibrated using a modified bisection method. For generation of the weekly forecasts, the model is fitted to the 7 day incidence of the new confirmed cases of the last 21 days including a nowcasting correction for the last week to supplement for subsequent registrations.

Results are gathered via Monte-Carlo simulations as the point-wise sample mean of multiple simulation runs. Due to the large number of agents in the model, eight simulation runs are used which are sufficient to have the sample mean approximate the real unknown mean with an error of less than 1% with 95% confidence (estimated by the Gaussian stopping as introduced in ref. [31]).

The model considers uncertainty with respect to the stochastic perturbations in the model by tracing the standard deviation of the Monte-Carlo simulations. Parameter uncertainty is considered in form of manually defined best and worst case scenarios.

**Epidemiological state space model**. The third model isolates weekday effects, statistical outliers and exogenous shocks in order accurately identify and extrapolate the current trend in reported new infections.

We therefore use a multivariate autoregressive state space model, where reported numbers of daily new infections (including corrections for exogenous shocks and nowcasting for the recent days) are explained by underlying latent factors ("states") and exogenous variables.

The basic structure of the model follows a random walk of order 2 and includes weekday effects that may change over time. Further information can be incorporated into the model by defining exogenous variables. The coefficients of these effects are not determined in the model but extracted from external sources. Currently, exogenous data included in the model comprise the effective immunization rate of the population, seasonal effects, number of imported cases, and NPI.

Effective immunization rates are calculated as the share of immunized individuals (vaccinated and/or recovered) times the corresponding empirically estimated effectiveness of protection against reported infections based on the screening method[32]. Seasonal forcing is modelled by the cosine function, with a maximum positive impact on transmission rates in January. The magnitude of seasonal effects are based on literature estimates[33]. The number of imported cases is based on Austrian contact-tracing data[34]. NPIs are incorporated based on information on NPIs in place at federal and regional level[35] and their reported effects on transmission rates[36].

For a more technical model description refer to Supplementary Note 2.

**Model harmonization**. In order to harmonize the model output and generate a single consolidated forecast for the number of new and accumulated positive COVID-19 tests, each model was set up to generate its output in a common data format for each of the nine federal states of Austria. Our forecasts consisted of time series of confirmed cases for each day starting with the number of positive tests at the day of the forecast consortium meeting at 11:59 pm and ending between 8 and 14 days in the future (over time, we slightly increased the forecast horizon).

Different averaging procedures were considered to generate the joined forecast from the three individual forecasts. These included the point-wise arithmetic and geometric mean as well as an adaptive weighting procedure wherein the timeseries for each

model contributes with weights proportional to the accuracy of its most recent forecasts. This strategy was adapted from previous work [24,37]. The following three strategies have been evaluated in terms of their forecast error. Let $F_i^j(t)$ denote the forecasts for the total number of COVID-19 cases on day $t$ for model $j$ for runs made in week $i$,

$$F_i^h(t) = f(F_i^1(t), F_i^2(t), F_i^3(t)) \qquad (1)$$

the harmonized forecast with strategy $f$, and $R_i$ the corresponding reported number.

- **Naive**. This strategy describes a static arithmetic average

$$f(F_i^1(t), F_i^2(t), F_i^3(t)) := \frac{1}{3}F_i^1(t) + \frac{1}{3}F_i^2(t) + \frac{1}{3}F_i^3(t) \qquad (2)$$

- **Geometric**. This strategy describes the static geometric average of the forecasts.

$$f(F_i^1(t), F_i^2(t), F_i^3(t)) := \sqrt[3]{F_i^1(t)F_i^2(t)F_i^3(t)} \qquad (3)$$

- **Continuously weighted dynamic mean**. This strategy describes a dynamically weighted arithmetic average.

$$f(x, y, z) := a_i^1 F_i^1(t) + a_i^2 F_i^2(t) + a_i^3 F_i^3(t) \qquad (4)$$

The weights are determined from the forecasting errors of the previous three weeks.

$$a_i^j = \frac{1}{3}\sum_{k=(i-3)}^{i-1} w_k^j, \quad w_k^j = \frac{\frac{1}{\max(|kjF - R_k|, 0.5)}}{\sum_{r=1}^3 \frac{1}{\max(|krF - R_k|, 0.5)}}, \; j \in \{1, 2, 3\}. \qquad (5)$$

**Hospital bed usage model**. Hospital occupancy is modelled in a stock-flow approach, in which the "stock" of hospital patients is changed over time by means of an in- and outflow of patients. Inflow (admission to ICU and normal wards) is calculated as a ratio of the time-delayed number of recently reported and projected new cases from the harmonized model forecast. Length of stay determines outflow. Admission rates are scaled in order to fit the current occupancy in all federal states. The scaling parameter (one for each federal state) can thus be interpreted as an effective hospitalization rate. Model parameters were initially extracted from literature[38] and subsequently calibrated to observational data to better fit the observed time series; see Supplementary Table 1 and Supplementary Table 2. Length of stay is modelled based on the empirical distribution of length of stay with a cut-off value of maximum 100 days. The hospital bed usage model is stratified by four age groups and sex as depicted in Supplementary Fig. 1. Mean average length of stay for admissions in the period January to May 2021 (and discharges in the period January to June 2021) was 13.4 for ICU stays and 10.7 for normal ward stays.

**Confidence intervals for forecasts**. Confidence intervals (CIs) for both case numbers of the harmonized model and hospital occupancy are derived from the empirical forecast error of each prognosis day. More specific, we retrospectively evaluate the ratio of the consolidated forecast and the observed seven-day-incidence rate of confirmed cases for each day over the forecast interval. The upper and lower limits of the 68% and 95% CIs used for the reporting of our forecasts are derived from the corresponding percentiles of the empirical distribution of the observed forecast error.

It is assumed that forecasting errors can be reasonably approximated by a log-normal distribution that is independent of the starting level of case numbers, that the variance increases with the forecast horizon, and that the forecast error of the first and second day of the forecast is skewed because increases due to delayed reporting are more likely than decreases in case numbers due to backdating. We therefore consider the relative logarithmic forecast error tuples $\Delta_{i,h}$ where $i$ denotes the number of the prognosis, and $h$ denotes the forecast horizon. These tuples are evaluated in a 2-dimensional kernel-density estimation (KDE). The resulting density function is evaluated at slices of $h = 1, 2, \ldots, H$, where $H$ is the last day of the forecast. 0.025, 0.16, 0.84 and 0.975 percentiles give thresholds for the error bands in the case number projections for $h$.

For the confidence intervals of the ICU and hospital occupancy we additionally take the occupancy on the first prognosis day into account because the fluctuations of the occupancy numbers played a much higher role for the error than the parameter uncertainty. Specifically, an almost linear relation between the level of uncertainty and the square root of the occupancy on the first prognosis day could be observed.

We regard the tuples $Y_{i,j} = (\Delta_{i,j}, \sqrt{X_i})$. Hereby, $\Delta_{i,j}$ denotes the logarithmic error $\log(X_{i,j}/\tilde{X}_{i,j})$ between the reported occupancy $X_{i,j}$ and the forecast occupancy $\tilde{X}_{i,j}$ of day $j$ in prognosis week $i$. Furthermore $X_i = X_{i,0} = \tilde{X}_{i,0}$ denotes the occupancy on the day of the prognosis in week $i$. The tuples $Y_{i,j}$ are evaluated in a 2-dimensional kernel-density estimation (KDE) using Scott's Rule to obtain the kernel function $f(\Delta, \sqrt{X})$.

To generate CIs, the marginal distributions

$$f_{\sqrt{X}}(\Delta) = \frac{f(\Delta, \sqrt{X})}{\int_\Delta f(u, \sqrt{X}) du}$$

are calculated numerically. Finally, the percentiles of $f_{\sqrt{X}}(\Delta)$ give the CIs for the logarithmic error, given a certain occupancy $X$ on the day of the prognosis.

**Forecasting error evaluation**. Forecasts are evaluated at the weekly official meetings of the consortium. A list of all dates (as well as officially reported forecasts) can be found online[23]. To quantify the error of the case forecasts, the total number of reported and projected new cases since the last meeting are compared using data as reported on the day of the meeting. For example, the forecast harmonized on April 3, 2020 was evaluated on the meeting on April 10th by comparing the total number of new cases reported between April 3rd and April 9th with the projected ones.

To avoid bias, relative and not absolute differences and errors are taken into account.

**Reporting summary**. Further information on research design is available in the Nature Portfolio Reporting Summary linked to this article.

## Results
**Forecasting positive test numbers**. We show the results for our rolling forecasts compared with the actual case numbers in Fig. 1 for 72 forecasts. For the time period from April 4 to Summer 2021, we performed and harmonized weekly forecasts that are visibly as bundles of lines in Fig. 1. The first published forecasts coincided closely with the peak of the first epidemic wave in Austria. While the models showed a clearly discernible divergence for the first forecast, the agreement increased over the first wave. The starting points for the early weekly forecasts occasionally lie below the actual cases due to a substantial amount of very late reporting of cases in these early periods of the epidemic. From May until July, reported cases remained at a comparably low level.

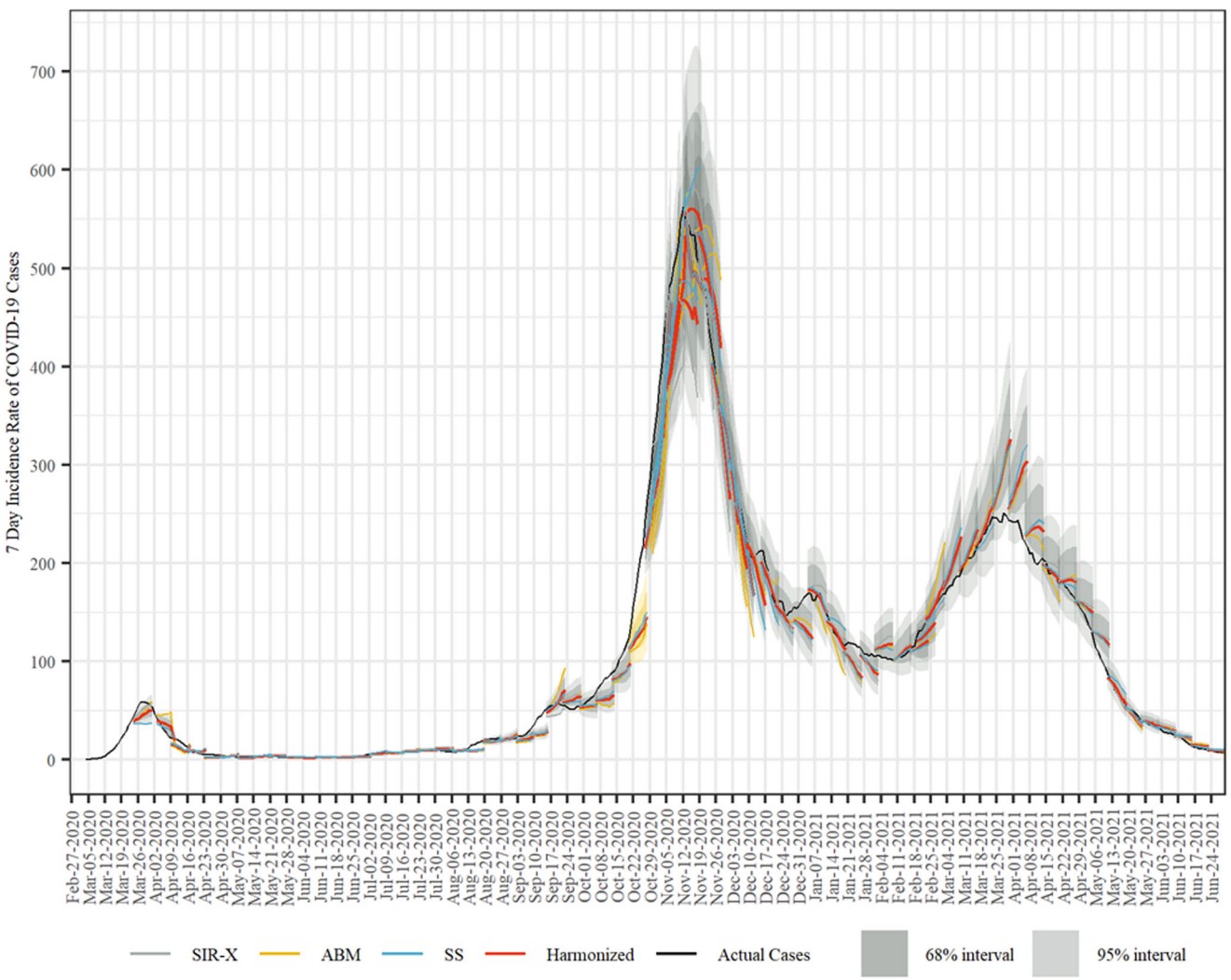

**Fig. 1 Rolling combined and consolidated out-of-sample forecasts for the 7 day incidence rate of confirmed cases in Austria.** We show the weekly predictions from the three different models, their arithmetic average with it corresponding CI, and the actual case numbers. The underlying data is found in Supplementary Data 1.

Starting from late August to early September, the onset of the second wave is visible. A critical point was the prognosis for the week until October 25, highlighted in yellow in Fig. 1. There, the consolidated forecast clearly underestimated the actual rise in confirmed cases. The forecasts made in the weeks thereafter were more accurate, however, the CI strongly increased due to the less accurate forecasts from the first half of October. A gradual flattening of the curve until end of January can be observed after the steep increase in November, during which the forecasts had a tendency to overestimate the degree of the flattening.

The early increase of the third wave in March 2021 was anticipated by the models due to the alpha variant becoming dominant. However, the peak of the third wave was not anticipated, as were the substantial decreases in reported cases in the following weeks when possibly seasonal factors increasingly curbed the virus spread.

In Fig. 2, we compare the model-specific forecast error with the forecast error of the consolidated model, the incidence and the effective reproduction number in Austria. In April 2020, agreement amongst the three models is typically stronger than the agreement with the data, meaning if one model over- or underestimated the actual trend, so did the other models. After the summer, agreement between the three models was larger than

in the early phases of the pandemic. Comparing the upper and lower part of this image also shows that none of the models anticipated a spontaneous rise in $R_{\mathrm{eff}}$ in combination with a large number of daily cases in August/September and October. This was particularly well visible at end of October (see the dotted lines in Fig. 2). Nevertheless, the violin representation in Fig. 3, which shows the relative differences between forecasts and reported data of all forecasts ever made, indicates that no persistent systematic over- or underestimation occurs while it cannot be ruled out that such systematic aberrations might have occurred over smaller periods of time.

We investigated the performance of different averaging procedures that weigh models according to their past performance in terms of their relative difference and error, see Methods. The results are summarized in Table 1. Performance weighting procedures yielded only a marginal improvement over simple averaging in terms of forecast accuracy. This further corroborates that agreement amongst the model forecasts is typically higher than their agreement with the observed case numbers.

**Forecasting bed usage**. In Fig. 4, we show our rolling forecasts for the number of intensive care beds in use for COVID-19

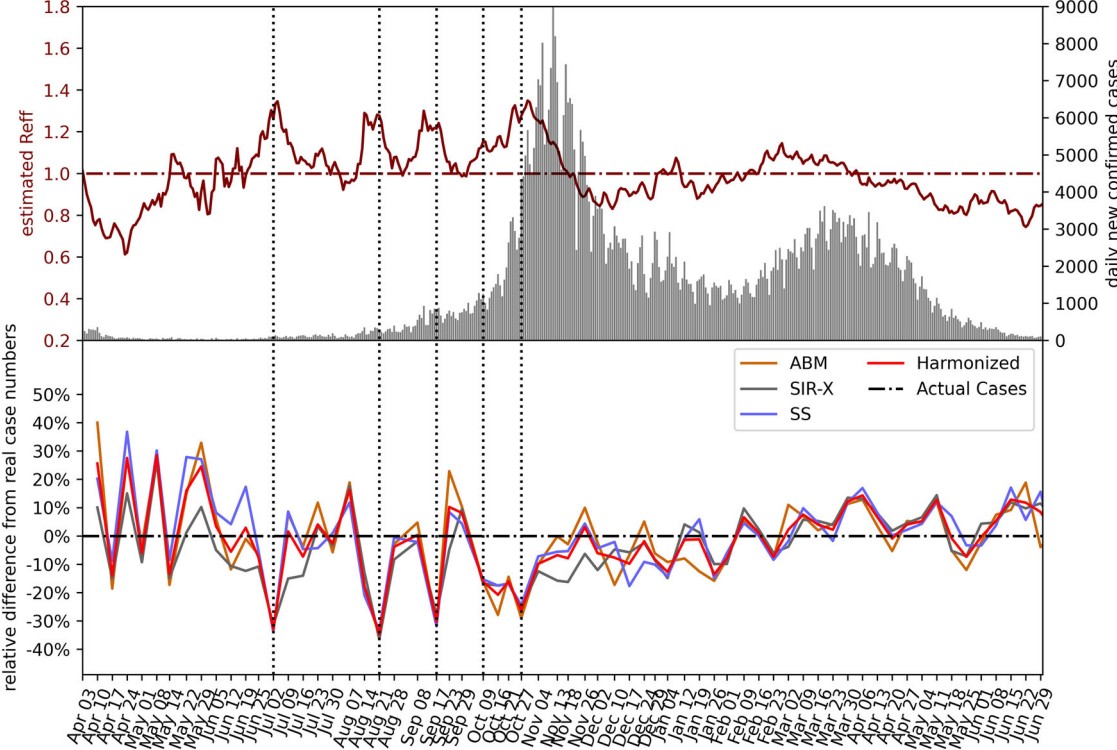

**Fig. 2 Evaluation of the forecast performance.** Daily new confirmed cases and estimated $R_{eff}$[9] (top panel) and forecasting performance (relative difference, bottom panel) of the individual models and the harmonized mean are shown for 72 forecasts. Weeks in which the forecasts substantially underestimated the actual case numbers (vertical dotted lines) tend to coincide with steep increases in $R_{eff}$. The underlying data is found in Supplementary Data 2.

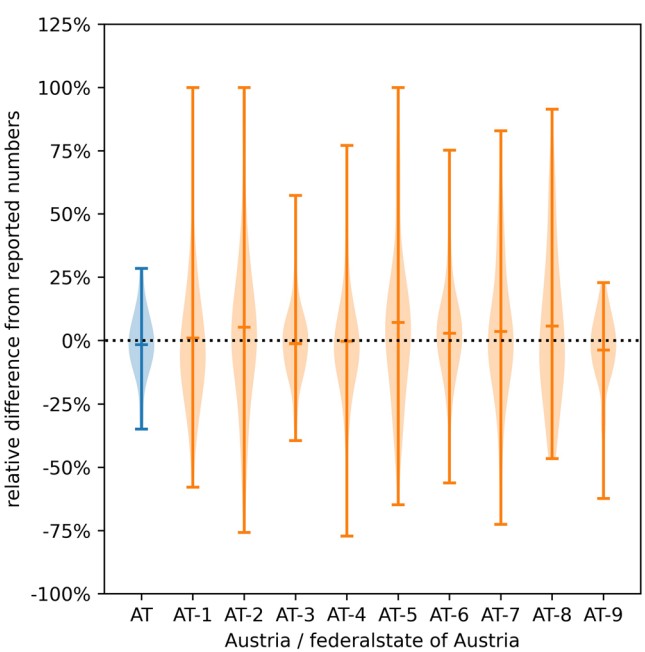

**Fig. 3 Evaluation of the forecasting performance for federal states.** The distribution of relative differences over all 72 forecasts to the reported data for Austria (AT) and all nine federal states (AT-1 to AT-9) is shown. Since the mean relative differences are close to zero, no persistent systematic over or underestimation of the cases occurs. For large federal states such as Lower Austria (AT-3) and Vienna (AT-9), forecasts are more reliable. The underlying data is found in Supplementary Data 3.

---

**Table 1 Forecast accuracy with different model harmonization strategies.**

Week 22, 2020-09-08 to 2020-09-17 (strong underestimate due to decline of seasonal effects)

| Strategy | Formula for mean (SIR-X (x), AB (y), SS (z)) | Relative error |
|---|---|---|
| Naive | $0.33x + 0.33y + 0.33z$ | 30.4% |
| Geometric | $\sqrt[3]{xyz}$ | 30.5% |
| Continuous | $0.36x + 0.32y + 0.32z$ | 30.7% |

Week 51, 2021-03-30 to 2021-04-06 (overestimated seasonality-caused turning point in spring)

| Strategy | Formula | Relative error |
|---|---|---|
| Naive | $0.33x + 0.33y + 0.33z$ | 14.3% |
| Geometric | $\sqrt[3]{xyz}$ | 14.3% |
| Continuous | $0.36x + 0.37y + 0.28z$ | 14.4% |

Week 45, 2021-02-01 to 2021-02-09 (well predicted turnaround of case numbers after lockdown end)

| Strategy | Formula | Relative error |
|---|---|---|
| Naive | $0.33x + 0.33y + 0.33z$ | 6.6% |
| Geometric | $\sqrt[3]{xyz}$ | 6.6% |
| Continuous | $0.20x + 0.36y + 0.44z$ | 7.3% |

Week 40, 2021-01-12 to 2021-01-19 (very well predicted decline under constant measures)

| Strategy | Formula | Relative error |
|---|---|---|
| Naive | $0.33x + 0.33y + 0.33z$ | 1.2% |
| Geometric | $\sqrt[3]{xyz}$ | 1.8% |
| Continuous | $0.76x + 0.08y + 0.16z$ | 0.6% |

Weeks 2–86

| Strategy | Median relative error | iqr |
|---|---|---|
| Naive | 5.21% | 9.70% |
| Geometric | 5.23% | 10.02% |
| Continuous | 4.98% | 10.06% |

We consider a naive arithmetic mean (strategy "naive"), geometric mean ("geometric") as well as an adaptive method ("continuous") which weights the models according to their recent performance. The table displays harmonization function as well as relative errors for selected weeks and a summary of the forecasts for weeks 2–86. None of the harmonization strategies stands out.

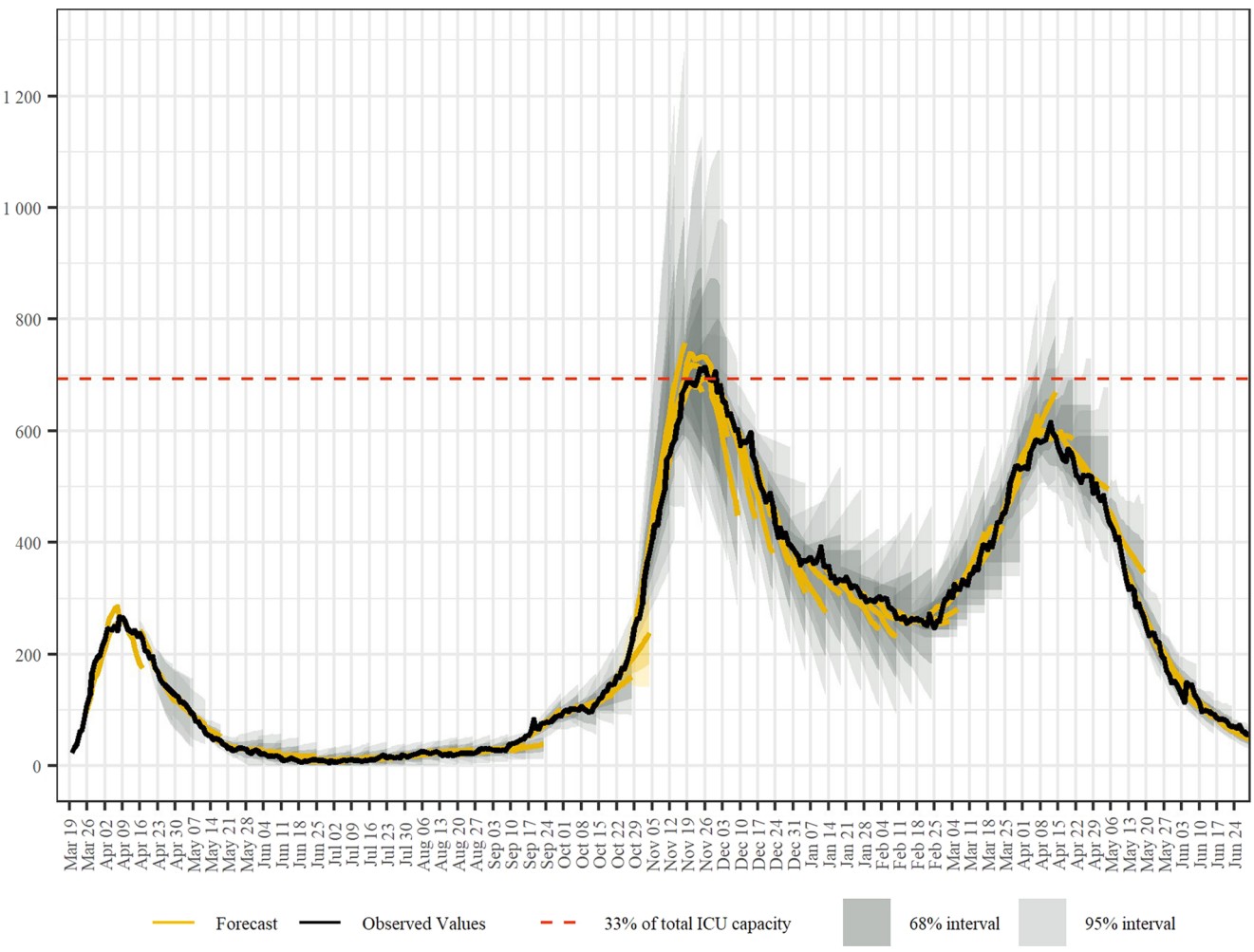

**Fig. 4 Forecasting bed usage.** Rolling combined and consolidated out-of-sample forecasts for the number of intensive care beds currently in use for COVID-19 patients including corresponding CI, the actual numbers of beds occupied, and 33% of total ICU capacity as a reference. The underlying data is found in Supplementary Data 4.

patients. There are three peaks in ICU bed occupancy corresponding and delayed with respect to the peaks in case numbers. The red dashed line gives the threshold of 33% of all ICU beds being in use for COVID-19 patients. We highlight again (yellow in Fig. 4) the forecast for the week until October 25, in which we severely underestimated the developments. After adjusting the models for these developments, the next forecast projected that the 33% threshold would likely be crossed in two to three weeks. After the second peak passed, the model showed a tendency to overestimate the pace at which patients would be released from ICU. Forecast accuracy during the third wave was overall much higher for the ICU forecast compared to the forecasts of the case numbers.

**Reporting of the forecasts**. We developed a standard reporting template to communicate our forecasts to other stakeholders and decision-makers, see Fig. 5. These visual reports showed our forecasts for cases and hospital occupancy, as well as information on the effective reproduction number. The visual reports are complemented by a brief synopsis of the researcher's appraisal of the current situation and particularities of the most recent forecast. These appraisals are publicly accessible [23]. Furthermore, the researchers are at disposal for any questions that members of the health ministers' office or the regional crisis management units may have.

The first panel in Fig. 5 provides an outlook for the expected developments in weekly cases (per 100,000 population). Due to known weekday effects, we do not give forecasts for daily case numbers. Expected hospital occupancy is given in the second panel. All forecasts are supplemented with 68% and 95% confidence intervals. The forecasts are also presented in the weekly sessions of the Austrian Corona Commission, an advisory body to the minister of health tasked with assessing the epidemiological risk in Austria. The expected occupancy rates of ICU is an indicator in assessing the risk of health system overburdening and thus directly contributes to the classification of Austrian regions according to this risk, which in turn informs recommendations on easing or strengthening control measures[6].

**Discussion**

Considering the impact of COVID-19 policy measures on economic and social life, any related decision support needs to be done with caution. Our approach considers the high impact of COVID-19 forecasts by (1) focusing on monitoring rather than long-term prognosis and (2) consolidation of three different "model opinions" which not only improved the quality of the short-term forecast, but also distributed the responsibility of the decision support on multiple teams.

From the very beginning of our work as consortium, we decided to publish only short-term forecasts and to refrain from

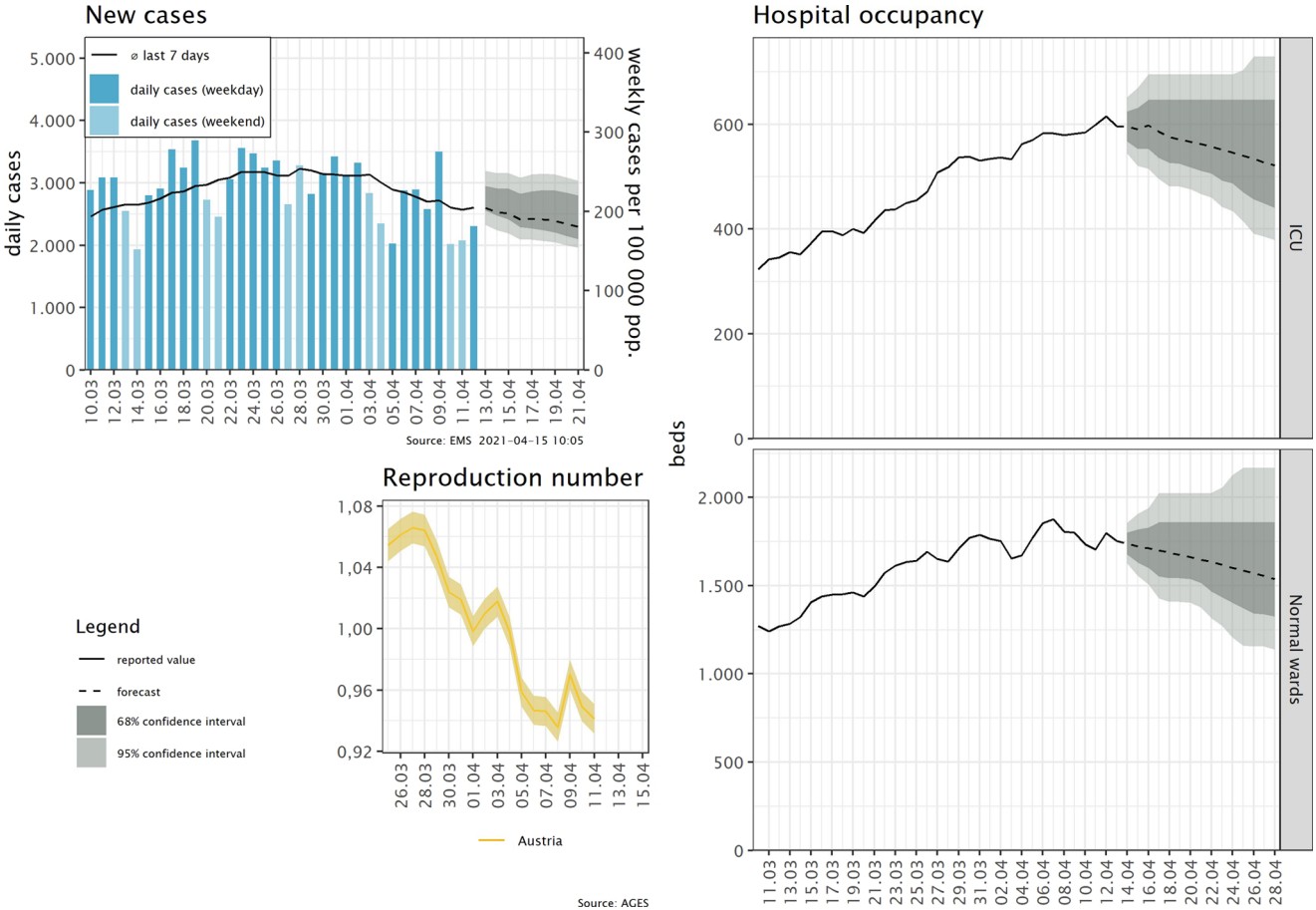

**Fig. 5 Example for a reporting template of our out-of-sample forecasts.** The visual reports consist of four panels. First, we report the daily number of cases in absolute terms and the weekly case numbers per 100 000 population as well as the forecasts for the weekly case numbers, including CIs. Additional panels show the the effective reproduction number and the forecasts for intensive and normal care beds occupancy with COVID-19 patients along with the corresponding CIs.

publishing longer-term scenarios. Due to the multiplicative growth of uncertainties in epidemiological models, accurate forecasts are typically only possible over a time horizon of several days[39–42]. There is no meaningful way to estimate the uncertainty for long-term scenarios that span over several weeks, months or even years. For policy-makers and non-technical experts, the distinction between a prognosis with a defined level of certainty and a hypothetical what-if experiment is hard to communicate.

Our forecasts provided evidence for the expected total number of daily infections and hospitalized cases including appraisals of uncertainty via forecast intervals. This is in contrast to what has been popularly described as "worst case coronavirus science", i.e. the communication of worst case scenarios as baselines in the public pandemic management strategy [43]. For instance, the UK policy change toward adopting more drastic NPIs on March 23 was based on worst case scenarios created by the Imperial College COVID-19 Response Team that within the current policy regime 250,000 deaths were to be expected. In a press conference in April, the Austrian chancellor publicly stated that soon "everyone will know someone who died because of COVID-19", based on an external SIR-model-based worst case scenario that contained a death toll of 100,000 people (1.1% of the Austrian population).

Such scenarios are problematic due to the high levels of uncertainty of long-term (multiple weeks and months) case number forecasts; a generic feature of mathematical epidemiological models which has been put under the spotlight by the COVID-19 pandemic[40,42,44,45]. Based on our results, we argue that a main

benefit of epidemiological models comes from their use as short-term monitoring systems. The models are typically calibrated to the infection dynamics of the last couple of days or weeks and forward project this dynamic based on epidemiological parameters often assumed to be fixed. If a short-term forecast is accurate this means that infection numbers have continued as expected, based on the recent trend and accurate assumptions. In our forecasts, as shown in Fig. 2, this holds true for some but not all time periods. If the short-term forecast severely over- or underestimates the observed dynamics, one should inquire more closely what might have caused this change.

Inaccurate short-term forecasts, therefore, signal a change in the epidemiological situation that needs to be explained. This occurred, for instance, when our forecasts in August 2020 consistently underestimated the observed case numbers. At this point, contact tracing data revealed a growing number of travel associated cases mainly from hotspots in South Eastern Europe contributed substantially to this unexpected increase in infections[34]. In response, novel border restrictions for persons entering Austria from these countries were put in place at the end of August. During the summer 2020, infection numbers increased from around 20 confirmed cases per day to about 200 cases, mostly driven by patients aged below 40 years. Consequently, the number of severe COVID-19 cases remained low and the effective rate to require intensive care dropped to one percent and below. The situation changed qualitatively in September, when not only case numbers started to soar again, but also hospital admissions increased much more strongly than

projected. Our analysis revealed that the driver behind the observed above-forecast ICU occupancy was above-forecast total case numbers, while age-specific ICU rates remained constant in our models. In other words, the bed usage forecasts were inaccurate because of the infection number forecast, but not because the characteristics of the detected cases changed in terms of severity (e.g., age or more symptomatic or severe cases).

At the onset of the second COVID-19 wave in Austria in October 2020 the forecasts influenced policy debate and contributed to the decision of implementing a second lockdown. The forecast of October 30 predicted an increase in ICU occupancy by COVID-19 patients from 263 to 681 (34% of total ICU capacity of 2,007 beds) within the next 15 days and stressed that capacity limits of around 700 to 800 ICU beds dedicated to COVID-19 patients may be exceeded by mid to end of November if this trend would be unbroken. Additionally, the heterogeneous trend across Austrian federal states – leading to regions with COVID-19 ICU occupancy rates of more than 50% – was emphasized[46]. In a speech in the Main Committee of the National Council the Austrian Federal Minister of Health emphasized the dynamic of the second wave. Referring to the forecast of October 30 which predicted an increase in daily case numbers up to 6,300 by November 7 and a critically high level of ICU occupancy rate, the Minister called for a second lockdown[46,47]. While actual ICU occupancy first remained below the forecast (e.g. 585 vs. 681 on November 14) the call for action turned out to be appropriate as the Austrian intensive care system operated around its capacity limit with a maximum of 709 ICU beds occupied by COVID-19 patients a bit later on November 25 according to available information.

After relaxing NPIs during December 2020 the forecasts also contributed to the third lockdown after Christmas where persistently high levels of ICU occupancy rates were predicted and concerns regarding the seasonal increase in contacts which may lead to a further increase in occupancy rates were raised[48].

In total, within the study period we observed three waves. The models anticipated the peaks or at least a substantial flattening of the curves in wave 1 and 2 but failed to anticipate the observed flattening of the third wave. The models also predicted the onset of the third wave while the early growth of the second wave was substantially stronger than forecasted. As discussed, the peaks of the first two waves where the result of hard lockdowns, while the third wave broke in several Austrian regions without a substantial strengthening of NPIs, most likely due to seasonal influences. This can also be seen in Fig. 2, where the effective reproduction number continued to decrease on May and June 2021 at a time where NPIs where gradually eased. The onset of the third wave, however, coincided with the takeover of the Alpha variant in Austria, which was successfully anticipated by the models. Taken together, these findings suggest that mechanistic epidemiological model can foresee certain types of turning points (e.g., due to NPIs or the emergence of more transmissible variants), while further research is needed to integrate other classes of drivers, such as changes in mobility[22] or meteorological factors[33].

One might question whether complex epidemiological models are necessary for such a short-term forecast-based monitoring system or whether the use of simpler models could not serve a similar purpose. Indeed, models that are not of the compartmental type but use other prediction algorithms, ranging from ARIMA[49,50] to deep learning[51,52], have also been applied to forecast the SARS-CoV-2 pandemic. The advantage of using compartmental epidemiological models is that they also provide a mechanistic description of why changes in the current epidemiological situation are occurring. In particular, as a consortium we were frequently asked by policy-makers to provide estimates for possible future epidemiological developments given a certain NPI would be implemented in a few days or not. To answer such requests, it is beneficial to use models allowing to directly simulate the effects of hypothetical interventions. Such questions can be even more reliably and consistently answered if the same model is used to produce short-term forecasts as a baseline scenario and a hypothetical scenario assuming the implementation of a new NPI.

Our forecast-based decision support comes with limitations. First, the weekly prognosis is partly based on shared data from the Ministry of Health and the Ministry of Internal Affairs which comes with quality and reporting bias limitations. Moreover, even though the consortium has access to the most accurate and up-to-date data about the epidemic in Austria, a lot of information required for valid epidemiological forecasting is not available or only available with considerable delay, since adequate reporting systems are not in place; e.g., the fraction of undetected persons due to asymptomatic disease progression. Further, our forecast is based on simulation models which are generally subject to errors that come from abstraction and simplification of the real system. Through the harmonized handling of three models with entirely different approaches we attempted to reduce such structural uncertainties. Finally, our decision support framework is mostly limited by its political and public visibility. According to our experience, our forecast was of special public and policy interest in periods of rapid movements but also had a confirmatory effect in times of decreasing case numbers or slow growth with respect to taken policy measures.

In conclusion, we argue that modellers need to be cautious and responsible in communicating the sometimes strongly speculative nature of their results and their uncertainties to politicians and the public. Short-term epidemiological models can be valuable ingredients of a comprehensive monitoring and reporting system to detect epidemiological change points, identify their potential causes, and thereby inform decisions to ease or strengthen governmental responses.

## Data availability

SARS-CoV-2 case and hopsitalization data for Austria is available via the COVID-19 open government data portal under https://www.data.gv.at/covid-19/. The data underlying Figs. 1 to 4 in this work can be found in Supplementary Data 1 to 4, respectively.

## Code availability

No custom code was written for data collection. We describe a reporting system that combines the output from several computational models which have been previously described in the literature; the Extended SIR-X Model[17], the Agent-Based SEIR Model[13] and the Epidemiological State Space Model[53].

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

## Acknowledgements

We thank Reinhild Strauss, Gabriela El Belazi, Lukas Richter, Daniela Schmid, Erich Neuwirth, and Uwe Siebert for helpful and stimulating discussions. Our work as COVID-19 Forecast Consortium was financially supported by the Federal Ministry for Social Affairs, Health, Care and Consumer Protection. The funders had no role in study design, data collection and analysis, decision to publish, or preparation of the manuscript.

## Author contributions

M.B., C.R., and N.P. developed and operated the Agent-Based SEIR Model. M.Z., L.R., F.B., and H.O. developed and operated the Epidemiological State Space Model. S.T. and P.K. developed and operated the Extended SIR-X Model. M.B., P.K., M.Z., L.R. and F.B. developed the hospital bed occupancy model. P.K. wrote the first draft of the article. M.B., M.Z., L.R., F.B. and C.R. contributed to the writing. All authors reviewed and edited the manuscript.

## Competing interests

The authors declare no competing interests.

## Ethical approval

Ethical approval and need for consent was waived in this modelling study that used publicly available data and aggregated information from the official Austrian COVID-19 disease reporting system.

**Additional information**

