## [Peer Review File · Communications Medicine]

Reviewers' comments:

Reviewer #1 (Remarks to the Author):

Bicher et al present a short-term forecasting system for Austrian covid response. They produce a framework by combining three different models and producing estimates for daily cases and hospitalizations. The models are calibrated to Austrian data and take into account the delays observed in Austrian data. I think work is very well motivated and a need of the hour.

However, the manuscript lacks details about the model and data. The model description is not adequate to really make it reproducible or understandable in its entirety. The description and details of the data used are almost not there. I think authors should give details so that the system is useful in the future or could be reused by others.

Another detail lacking is how exactly CIs are calculated for the ensemble model. Also, authors have shown their estimates and in passing suggest other models that have been used in literature. A comparison with at least 1/2 of other work is required to get an idea about why models suggested by authors are better or more suited.

Reviewer #2 (Remarks to the Author):

In this manuscript, the authors provide a summary of methodologies used to inform Austrian governments on the burden of COVID-19. In particular, they are/were interested in estimating if a certain threshold for ICU occupancy will be reached due to emergence of new waves. Their manuscript aims to communicate various modelling methodologies (and their differences) and the degree of success they had. Consequently, a key strength of their study is highlighting the differences in results/forecasting due to model structure and assumptions.

I am recommending this manuscript for publication, though I have some minor comments:

- How is the uncertainty of parameters in the agent-based model implemented? It is unclear to me whether some or all disease-specific parameters follow statistical distributions or were they all fixed estimates? I am talking about parameters such as the incubation period, infectiousness period, etc.
- Is it correct to think that model forecasts were of "confirmed cases" rather than infections? I am not sure how a case can be confirmed when forecasting in the future. Maybe the language surrounding these results needs to be clarified.
- Speaking about confirmed cases, could these models be used to infer the true number of infections in Austria, including asymptomatic infections or infections that were missed due to lack of testing? That might be a very interesting statistic for estimating the herd immunity in the country.
- Could the authors clarify how exactly they calculated confidence intervals? Might they be calculating a range of values (that is what is suggested by the sentence: "The upper and lower limits of the CI are derived from the corresponding percentiles of the empirical distribution of this forecast error")

Typo: line 252

Reviewer #3 (Remarks to the Author):

In the manuscript "Supporting COVID-19 Policy-Making with a Predictive Epidemiological Multi-Model Warning System", the authors reflect on their work in the Austrian COVID-19 forecast

consortium. They briefly present the methods they deployed to evaluate developments in the local infection dynamics and to inform the Austrian government to facilitate policy decisions regarding the public health situation. The manuscript describes a consolidated modeling-based forecast regarding the cumulative case counts, as well as hospital bed and ICU bed occupancy. The consolidated forecast consists of independent contributions of three models: an ODE-based compartmental infectious disease model, an agent-based infectious disease model, and a state-space model. In addition to describing the process of developing a forecasting procedure, the authors discuss their role as scientists in a governmental advisory position.

While the article is of interest to a broad audience and provides relevant insights into opportunities and pitfalls of science communication it mostly evolves around reporting experiences and describing the general *modus operandi* of the Austrian COVID-19 taskforce. It thus forms an interesting and worthwhile perspective on how scientific evidence can inform public policy but simultaneously lacks the depth, scientific novelty and reproducibility that is required for an original research article. The authors are hence invited to resubmit their work as a perspective paper after addressing the below comments.

Major comments:

* Lines 245-263: The model description remains very vague and partly differs from the SI/Appendix B. For example, the main text states that Kalman Filtering is used but in the SI a package is referred to that does forecasts based on an ARIMA. Even with the appendix, reproducibility seems to pose a difficult task (see minor comments section). Please be clear on what was done, for instance regarding the sentence "Uncertainty can be modelled by varying underlying model parameters[...]", can uncertainty be modelled like that or was it indeed modelled like that, and if so, which parameters were varied (there is no further mention of this in the SI)?. Regarding the SI, the whole section B.1 is written in a relatively broad and ambiguous manner such that reproducibility is hindered. In section B.2 model parameters are defined using mathematical symbols that are not used again in section B.3. It is unclear what "infection rate" is modelled. "Infection rate" can refer to a multitude of things in the context of infectious disease modeling, e.g., individual infection probability per unit time per contact, mean-field infection probability per unit time, or simply the temporal derivative of the logarithm of the incidence. Furthermore, it is difficult to understand for which part of the model which type of input data is used. It might help to define quantities using symbols instead of words and then clearly define which symbols refer to data and which refer to latent quantities. E.g. defining an observable as "background infection risk_t = population/100,000" is less clear than the authors might think: it has not been defined what "population" means, there is no temporal dependence on the RHS of this equation even though there is one on the LHS and "background infection risk" can refer to many things. Please provide a thorough mathematical description of the model with clear definitions of the quantities used.

* Line 280: The discrete averaging seems very arbitrary and it adds unnecessary complexity to the paper. I would suggest removing it unless you could provide the reader with some references on why the proposed approach works better in certain situations.

* Line 328: Does the prediction improve because the models get better over time or because the case numbers are low during those periods? A careful reader might suspect the latter.

* Table 1: The authors are invited to show weeks with large prediction errors as well. One may only assume that errors become significantly larger at the onset and turning points of waves (this is also clearly visible in Figure 1 and discussed in a later section of the paper). Such errors should be quantified and discussed.

* It is suggested to show and discuss Figure 1 for New Cases/Incidences instead of cumulative case counts. The latter doesn't play a role for ICU occupancies and makes it more difficult to visually inspect differences between observed and predicted numbers.

* Along those lines, it remains unclear which relative error is shown in Figure 2. Potential candidates are the relative error between daily new cases (or a 7d average of daily new cases) or the relative error of the total number of cases. When comparing the forecast of cumulative case numbers around Oct 15, 2020 (shown in Figure 1) with the actual data, the corresponding relative error shown in Figure 2 seems to be in agreement. However, since the cumulative number of confirmed cases has no relevance for the dynamic evaluation of the public health situation, the

relative error between predicted and actual new cases (i.e. incidence) should be used instead. Otherwise, with growing cumulative confirmed cases, the relative error of forecast and data will naturally become smaller over time, thereby artificially increasing the forecast's apparent accuracy. Additionally, it is unclear which temporal difference was chosen between the date *on which* the forecast was made and the date *for which* the forecast was made to compute the relative error in Figure 2. Do the authors show the relative difference on a weekly basis (i.e. time delay of 7d)?

* Line 472: The forecasts over- or underestimate observed dynamics mostly around the onsets or turning points of waves. The authors argue in the lines before, that forecasts are accurate if the numbers continue as expected. Exactly this behaviour is not true for onsets and turning points. So one might ask somewhat provocatively why the authors employ such sophisticated models in the first place instead of simply extrapolating trends given that they acknowledge that their models are also unable to capture the most interesting/important parts of the dynamics. A partial answer is given in Line 544, but the authors should elaborate further on this important subject to justify their approach to critical readers.

* Forecasts should ideally have a mean relative error of zero after quite some time, i.e. differences between forecast and data should be distributed symmetrically around zero. Figure 2 indicates that this is not the case in the present setting but that the forecasts consistently underestimate the data. This implies a strong systematic bias in the forecast procedure which should be discussed and explained thoroughly in the revised paper. The same holds for the ICU forecasts which mostly predict lower numbers than provided by the actual data (see e.g. after the peak in November).

Minor comments and clarifying questions:

* Line 53: Rephrase "remove piling bodies"

* Line 103: Who is meant by "our" -- the authors or the "The Austrian Corona Commission". If the reference is with respect to the latter, one should explain the members of the Commission already at line 65.

* Provide references to the employed models already in Line 154 to 160, since the paper otherwise creates the impression that the models were developed (and not used or extended) by the institutions in brackets

* Line 209: Which sampling methods are used and what is the number of representative agents (compared to the Austrian population)?

* Line 224: Which NPIs were modelled and how do they affect the simulation outcomes?

* Line 226: What is the number of parameters in the model? Since these numbers are typically quite large in ABMs, what measures are taken to avoid overfitting?

* Line 245: Please clarify what a state space model is.

* Line 272: Remove one "and"

* Line 284-286: "Until the end of September, CIs were derived from the SIR-X... using the empirical ... error of the harmonized model." This sentence needs rewriting. Are the CIs derived from the SIR-X or from harmonized model? And why was this procedure only applied until the end of September?

* Table 1: Please use respective dates instead of giving week numbers in order to make the table comparable with Figure 1

* Table 1: How exactly are the errors computed? Is it the average error over all days of a week, or the error at a specific forecast horizon?

* Line 316: The SI is lacking a collection of model parameters, please provide one.

* Line 441: Please provide a reference for this rather fundamental form of criticism (one example does not suffice here)

* Line 480: Please provide a reference for this statement

* Line 819: "[...] based on the cluster analysis." Which cluster analysis? Please add a reference.

* Line 826: "We use exponential smoothing to identify seasonality, error and level of case numbers [...]" please specify what you mean with identify. Do you mean that you remove seasonality and error with the exponential smoothing?

* Line 846: "standard case, there would be [...]" What is meant by standard? Please add a reference here.

* Line 849, Eq. B4: change "total infectious"->"total infectious cases_t"

* Line 860: "may err as well", please rewrite

- * Line 787: The outflux " $-\kappa_0 I$ " is not reflected by any proportional influx term for any other compartments. Should Eq. A8 be equal to Eq. A4?
- * It remains unclear how the authors computed R_{eff} in Figure 2. There are several methods to compute R_{eff} but none have been cited and/or explained
- * line 459: replace "outbreak" with "pandemic"
- * line 830: Please clarify what is meant by "increase the number of infectious.." -- this sentence seems incomplete

We thank the reviewers for their thorough reading of our manuscript and for making valuable suggestions for how to further improve the quality of the work. We have taken each of these comments thoroughly into account and revised the manuscript accordingly. In particular, we have ensured that the technical description is now substantially more detailed as requested. Below we give a point by point response and hope that the manuscript has now become acceptable for publication.

Reviewer #1 (Remarks to the Author):

Bicher et al present a short-term forecasting system for Austrian covid response. They produce a framework by combining three different models and producing estimates for daily cases and hospitalizations. The models are calibrated to Austrian data and take into account the delays observed in Austrian data. I think work is very well motivated and a need of the hour.

However, the manuscript lacks details about the model and data. The model description is not adequate to really make it reproducible or understandable in its entirety. The description and details of the data used are almost not there. I think authors should give details so that the system is useful in the future or could be reused by others.

We thank the reviewer for commenting positively on the necessity of our work and for bringing to our attention that the manuscript did not communicate clearly wherein the main technical novelty of our approach lies. Concerning a description of the model and the data, note that the main novelty of our work does not lie in the development of a new modeling framework. Rather, we leveraged existing models, tailored them to available Austrian data and combined them into a multi-model short-term forecasting system as described here, including a forecasting model for hospital bed usage and properly defined forecast intervals. That is why the description of the models was rather barebone in the original manuscript, as we just summarized there previous work or how we adapted these models. We have now added a section in the introduction where this is now explicitly stated.

We now also provide more details on how the data was used in the Methods section, as requested.

Another detail lacking is how exactly CIs are calculated for the ensemble model. Also, authors have shown their estimates and in passing suggest other models that have been used in literature. A comparison with at least 1/2 of other work is required to get an idea about why models suggested by authors are better or more suited.

We added a more detailed description of the calculation of CIs.

We added 2 references that support the benefits of using consolidated harmonized forecasts of multiple models over using results from one single model.

Reviewer #2 (Remarks to the Author):

In this manuscript, the authors provide a summary of methodologies used to inform Austrian governments on the burden of COVID-19. In particular, they are/were interested in estimating if a certain threshold for ICU occupancy will be reached due to emergence of new waves. Their manuscript aims to communicate various modelling methodologies (and their differences) and the degree of success they had. Consequently, a key strength of their study is highlighting the differences in results/forecasting due to model structure and assumptions.

I am recommending this manuscript for publication, though I have some minor comments:

- How is the uncertainty of parameters in the agent-based model implemented? It is unclear to me whether some or all disease-specific parameters follow statistical distributions or were they all fixed estimates? I am talking about parameters such as the incubation period, infectiousness period, etc.

→ *Almost all disease durations are modeled as distributions with estimated moments. I.e. in case the simulation needs to specify how long a model-agent remains in a certain state, a pseudo-random-number-generator draws a number according to the specified distribution.*

In specific, a scaled Beta distribution is used for incubation time, a Gamma distribution is used for sampling the duration between symptom-onset and quarantine. A Triangular distribution is used for sampling recovery durations. Hospitalization durations (normal and ICU bed) are sampled from empirical distributions gathered from hospital records.

Anyway, we did not intend to specify the agent-based model in such a high level of detail in the present work (this would not be possible anyhow). Instead we would refer to already published and publicly available work about this specific model. We already did this in the original version of the ms, but gave it a stronger emphasis in the revised version.

- Is it correct to think that model forecasts were of "confirmed cases" rather than infections? I am not sure how a case can be confirmed when forecasting in the future. Maybe the language surrounding these results needs to be clarified.

→ *Yes, the agent-based model includes an estimate for an infected person/agent to remain undetected. So, the used outcome "confirmed-cases", which is used for calibration of the model to real data, only refers to a small part of all infections.*

- Speaking about confirmed cases, could these models be used to infer the true number of infections in Austria, including asymptomatic infections or infections that were missed due to lack of testing? That might be a very interesting statistic for estimating the herd immunity in the country.

→ *Indeed, studies with the ABM have been made that use this feature to get insights into herd-immunity (see <https://bmcinfectdis.biomedcentral.com/articles/10.1186/s12879-020-05737-6> , and (preprint) <https://www.medrxiv.org/content/10.1101/2021.03.10.21253251v1>)*

- Could the authors clarify how exactly they calculated confidence intervals? Might they be calculating a range of values (that is what is suggested by the sentence: "The upper and lower limits of the CI are derived from the corresponding percentiles of the empirical distribution of this forecast error")

We added a more detailed description of the calculation of CIs.

Typo: line 252

Thank you for spotting this typo!

Reviewer #3 (Remarks to the Author):

In the manuscript "Supporting COVID-19 Policy-Making with a Predictive Epidemiological Multi-Model Warning System", the authors reflect on their work in the Austrian COVID-19 forecast consortium. They briefly present the methods they deployed to evaluate developments in the local infection dynamics and to inform the Austrian government to facilitate policy decisions regarding the public health situation. The manuscript describes a consolidated modeling-based forecast regarding the cumulative case counts, as well as hospital bed and ICU bed occupancy. The consolidated forecast consists of independent contributions of three models: an ODE-based compartmental infectious disease model, an agent-based infectious disease model, and a state-space model. In addition to describing the process of developing a forecasting procedure, the authors discuss their role as scientists in a governmental advisory position.

While the article is of interest to a broad audience and provides relevant insights into opportunities and pitfalls of science communication it mostly evolves around reporting experiences and describing the general modus operandi of the Austrian COVID-19 taskforce. It thus forms an interesting and worthwhile perspective on how scientific evidence can inform public policy but simultaneously lacks the depth, scientific novelty and reproducibility that is required for an original research article. The authors are hence invited to resubmit their work as a perspective paper after addressing the below comments.

Major comments:

* Lines 245-263: The model description remains very vague and partly differs from the SI/Appendix B. For example, the main text states that Kalman Filtering is used but in the SI a package is referred to that does forecasts based on an ARIMA. Even with the appendix, reproducibility seems to pose a difficult task (see minor comments section). Please be clear on what was done, for instance regarding the sentence “Uncertainty can be modelled by varying underlying model parameters[...]”, can uncertainty be modelled like that or was it indeed modelled like that, and if so, which parameters were varied (there is no further mention of this in the SI)? Regarding the SI, the whole section B.1 is written in a relatively broad and ambiguous manner such that reproducibility is hindered. In section B.2 model parameters are defined using mathematical symbols that are not used again in section B.3. It is unclear what “infection rate” is modelled. “Infection rate” can refer to a

multitude of things in the context of infectious disease modeling, e.g., individual infection probability per unit time per contact, mean-field infection probability per unit time, or simply the temporal derivative of the logarithm of the incidence. Furthermore, it is difficult to understand for which part of the model which type of input data is used. It might help to define quantities using symbols instead of words and then clearly define which symbols refer to data and which refer to latent quantities. E.g. defining an observable as “background infection risk_t = population/100,000” is less clear than the authors might think: it has not been defined what “population” means, there is no temporal dependence on the RHS of this equation even though there is one on the LHS and “background infection risk” can refer to many things. Please provide a thorough mathematical description of the model with clear definitions of the quantities used.

Thank you very much for your comment. We completely revised our model description in the main text and in the SI in order to make our model reproducible. Therefore we improved the mathematical description of the model, precisely defined the quantities used as model inputs and included references to the R packages used.

* Line 280: The discrete averaging seems very arbitrary and it adds unnecessary complexity to the paper. I would suggest removing it unless you could provide the reader with some references on why the proposed approach works better in certain situations.

→ *We exchanged the ranking-based “discrete” averaging method by the geometric-mean of the forecasts. Moreover, we added a reference for the adaptive averaging concept.*

* Line 328: Does the prediction improve because the models get better over time or because the case numbers are low during those periods? A careful reader might suspect the latter.

→ *We updated Figure 2 by more recent forecasts and updated and revised data from the epidemiological reporting system. Also in the old, but more emphasised in the newer version of the plot, what can be seen is the following: the lower the case numbers the higher the relative differences of the forecasts due to the higher impact of hard-to-predict sporadic clusters (eg, some of the larger deviations in summer*

2020 were driven by clusters in slaughterhouses or in churches when in the weeks before the districts were close to zero cases, leading to large relative errors).

* Table 1: The authors are invited to show weeks with large prediction errors as well. One may only assume that errors become significantly larger at the onset and turning points of waves (this is also clearly visible in Figure 1 and discussed in a later section of the paper). Such errors should be quantified and discussed.

→ *We changed the strategy from showing more less arbitrary weeks (10,20,30) to displaying certain relevant weeks that support the Discussion section.*

* It is suggested to show and discuss Figure 1 for New Cases/Incidences instead of cumulative case counts. The latter doesn't play a role for ICU occupancies and makes it more difficult to visually inspect differences between observed and predicted numbers.

Figure 1 has been changed accordingly.

* Along those lines, it remains unclear which relative error is shown in Figure 2. Potential candidates are the relative error between daily new cases (or a 7d average of daily new cases) or the relative error of the total number of cases. When comparing the forecast of cumulative case numbers around Oct 15, 2020 (shown in Figure 1) with the actual data, the corresponding relative error shown in Figure 2 seems to be in agreement. However, since the cumulative number of confirmed cases has no relevance for the dynamic evaluation of the public health situation, the relative error between predicted and actual new cases (i.e. incidence) should be used instead. Otherwise, with growing cumulative confirmed cases, the relative error of forecast and data will naturally become smaller over time, thereby artificially increasing the forecast's apparent accuracy. Additionally, it is unclear which temporal difference was chosen between the date *on which* the forecast was made and the date *for which* the forecast was made to compute the relative error in Figure 2. Do the authors show the relative difference on a weekly basis (i.e. time delay of 7d)?

→ *In this evaluation we defined the error between the forecast and the official reporting system, by comparing the cumulative number of confirmed cases within the time between the forecasting meeting in which the prognosis was harmonized and the following forecasting meeting. Since forecasts are usually done weekly (with exceptions due to national holidays etc.) this mostly corresponds with the week-incidence.*

The error is now described more carefully in the main text.

* Line 472: The forecasts over- or underestimate observed dynamics mostly around the onsets or turning points of waves. The authors argue in the lines before, that forecasts are accurate if the numbers continue as expected. Exactly this behaviour is not true for onsets and turning points. So one might ask somewhat provocatively why the authors employ such sophisticated models in the first place instead of simply extrapolating trends given that they acknowledge that their models are also unable to capture the most interesting/important parts of the dynamics. A partial answer is given in Line 544, but the authors should elaborate further on this important subject to justify their approach to critical readers.

Actually, some of the turning points were to some extent anticipated because they were foreseeable due to specific factors. The peak (or at least a substantial flattening of the curve) of the first and second wave was expected by the models. This is obviously not true of the third wave. The onset of the third wave was anticipated while the models did not predict the onset of the second wave.

The turning points of waves 1 and 2 were the result of NPIs (hard lockdowns) which could be modelled. However, the third wave started to break also due to seasonal influences in several Austrian regions without substantial changes in NPIs (similar episodes played out eg in Germany), which the models did not anticipate. We also attribute the onset of the second wave to such seasonal influences (the onset was preceded by a substantial shift in weather from summer to autumn-like conditions) whereas the onset of the third wave coincides with the takeover of the Alpha variant, which could be anticipated in the models by increasing transmissibility. We discuss this now in more detail in the manuscript and use this to illustrate the advantages of using mechanistic models.

* Forecasts should ideally have a mean relative error of zero after quite some time, i.e. differences between forecast and data should be distributed symmetrically around zero. Figure 2 indicates that this is not the case in the present setting but that the forecasts consistently underestimate the data. This implies a strong systematic bias in the forecast procedure which should be discussed and explained thoroughly in the revised paper. The same holds for the ICU forecasts which mostly predict lower numbers than provided by the actual data (see e.g. after the peak in November).

→ *Indeed, the swift increase of case numbers in autumn 2020 caused several forecasts in which the case numbers were underestimated. Updating Figure 2 with newer data until July 2021 shows that this is not a consistent bias of the forecasts, but was caused by the development of the case numbers in autumn. For example, during the decline of the third wave in April 2021 the forecasts rather overestimated the case numbers.*

We added a violin plot of all relative differences between forecasts and reported numbers, which displays that no systematic under or overestimation occurs.

Minor comments and clarifying questions:

* Line 53: Rephrase “remove piling bodies”

Rephrased.

* Line 103: Who is meant by “our” -- the authors or the “The Austrian Corona Commission”. If the reference is with respect to the latter, one should explain the members of the Commission already at line 65.

* Provide references to the employed models already in Line 154 to 160, since the paper otherwise creates the impression that the models were developed (and not used or extended) by the institutions in brackets

* Line 209: Which sampling methods are used and what is the number of representative agents (compared to the Austrian population)?

→ *The agent-based model makes use of one statistical representative for each inhabitant of Austria (so, the model uses ~9 Mio agents). Every simulation run, these agents are randomly sampled according to given distributions for age, sex, and regional distribution. They are furthermore randomly assigned to locations (i.e. households, workplaces, and schoolclasses) given distributions for the sizes and age-structure of the corresponding location type.*

Anyway, we did not intend to specify the agent-based model in such a high level of detail in the present work (this would not be possible anyhow). Instead we would refer to already published and publicly available work about this specific model. We already did this in the original version of the ms, but gave it a stronger emphasis in the revised version.

* Line 224: Which NPIs were modelled and how do they affect the simulation outcomes?

→ *The agent-based model is capable of directly depicting various NPIs such as closure of workplaces, schools, and contact tracing. A proper analysis of the latter can be found in <https://journals.sagepub.com/doi/pdf/10.1177/0272989X211013306>. Clearly, there are several NPIs which cannot even be modeled directly with the ABM, such as face-mask-wearing or media campaigns. In these cases, the ABM is additionally calibrated using a general infectivity parameter and simply forecasts a trend.*

For all three models, sensitivity analysis of relevant parameters (which includes parameters of modelled NPIs) were performed, yet, including the results would exceed the scope of the present work.

* Line 226: What is the number of parameters in the model? Since these numbers are typically quite large in ABMs, what measures are taken to avoid overfitting?

→ *The ABM was developed under full consideration of the ISPOR-SMDM modeling good research practices. Therefore, the number of parameters (around 25) was chosen as small as possible, but as large as required to answer the original research problem (which goes far beyond the use of the model for short-term prognosis). Moreover, any parameter's value is defined and validated with two independent data sources and analyzed with proper sensitivity analysis. For more information on validation of the model, we refer to <https://doi.org/10.1177/0272989X211013306>.*

Overfitting of the forecasts is naturally avoided, since we calibrate the model not with all, but only one free scalar parameter.

* Line 245: Please clarify what a state space model is.

We added a description

* Line 272: Remove one “and”

Thank you for spotting this typo!

* Line 284-286: “Until the end of September, CIs were derived from the SIR-X.... using the empirical ... error of the harmonized model.” This sentence needs rewriting. Are the CIs derived from the SIR-X or from harmonized model? And why was this procedure only applied until the end of September?

At this point we had accumulated enough forecasts such that we could estimate our empirical forecast error in a statistically reliable way. Before that we used different techniques to obtain the CIs, e.g., by using CIs derived from one of the model. However, we acknowledge that it goes beyond the scope of this work to give a historical overview of which method was used at which point in time (particularly in the early days where everything was under heavy development) but instead consistently report the model formulations as they are currently used (and have in practice been used for most of the study period). We have therefore removed the statement referenced above, to avoid such confusions in the future.

* Table 1: Please use respective dates instead of giving week numbers in order to make the table comparable with Figure 1

→ *Done*

* Table 1: How exactly are the errors computed? Is it the average error over all days of a week, or the error at a specific forecast horizon?

→ *Added a small section in the Methods*

* Line 316: The SI is lacking a collection of model parameters, please provide one.

→ A section with tables and graphs including the model parameters (hospitalisation rates, distribution of length of stay, etc.) was added to the SI

* Line 441: Please provide a reference for this rather fundamental form of criticism (one example does not suffice here)

Added reference and rephrased.

* Line 480: Please provide a reference for this statement

Reference (Austrian Agency for Health and Food Safety) was added.

* Line 819: “[...] based on the cluster analysis.” Which cluster analysis? Please add a reference.

We added a Reference to the cluster analysis of the Austrian Agency for Health and Food Safety.

* Line 826: “We use exponential smoothing to identify seasonality, error and level of case numbers [...]” please specify what you mean with identify. Do you mean that you remove seasonality and error with the exponential smoothing?

* Line 846: “standard case, there would be [...]” What is meant by standard? Please add a reference here.

* Line 849, Eq. B4: change “total infectious”->“total infectious cases_t”

We completely revised the model description in the annex in order to enhance reproducibility

* Line 860: “may err as well”, please rewrite

Thank you for spotting this typo!

* Line 787: The outflux “- $\kappa_0 I$ ” is not reflected by any proportional influx term for any other compartments. Should Eq. A8 be equal to Eq. A4?

Thank you for spotting this typo!

* It remains unclear how the authors computed R_{eff} in Figure 2. There are several methods to compute R_{eff} but none have been cited and/or explained

Reference (AGES, Richter et al.) was added to caption of figure 2

* line 459: replace “outbreak” with “pandemic”

The wording has been changed accordingly

* line 830: Please clarify what is meant by “increase the number of infectious..” -- this sentence seems incomplete

We clarified the sentence by explaining the influence of imported cases.

Reviewers' comments:

Reviewer #2 (Remarks to the Author):

The revised manuscript is now more clear and provides necessary details.

Reviewer #3 (Remarks to the Author):

The authors improved the description of their methods and mostly replied to the raised comments. I applaud their efforts as advisors for the Austrian government regarding the country's response to the COVID-19 pandemic. Continuously consolidating forecasts obtained by three rather distinct methods is certainly a valuable effort to minimize critical errors in short-term science-informed consulting of legislation. Nevertheless, the initial impression that this manuscript is more suited as a commentary or perspective paper rather than an original research paper prevails. Each of the models seems very involved and it simply does not suffice to write four pages of rough descriptions of the respective models' workings in the Supplementary Information, without any sensitivity analyses, schematic explanations, data definitions. As an example for such shortcomings, in the SI Sec A.2 the authors briefly mention that they extended their original model to incorporate "age structure" in "the usual (sic) way", yet they do not cite any method for what "usual way" means. I suspect I know what they mean, but they do not explicitly mention the equations that they integrated, they do not explicitly provide the contact data they used, what age stratas they decided to use. They also mention that "the parameter alpha becomes a matrix", yet falsely attribute the term "likelihood" to a rate and write that the term α_{ij} is the likelihood that a susceptible of age group j will be infected by an infected from age group i . This is not a sufficient description that enables the reader to judge the validity of the model structure, as the matrix α_{ij} has to meet certain normalization conditions and there's no way for me to check that it does. Were the authors to use canonical contact structure data as provided by the POLYMOD or COVIMOD studies, or would have cited sources that explained how they built age structure into their model (for example with the R package `socialmixr`), this might be a different story, yet they simply state that they use "mobile phone data", I assume (they don't state it) implicitly setting a number of phone calls between age groups to be proportional to the number of epidemiologically relevant face-to-face contacts between age groups and deliver no justification for this procedure, which is, however, needed, because phone calls are not face-to-face contacts. They also do not explain the mobile phone data which might not be representative of the population structure. All of this might potentially lead to very skewed contact distributions and therefore skewed results, but there is simply no way of telling, even after the revision. On top of all that, the authors never mention why exactly they incorporate age structure in the first place and how it improves their method as compared to the simple model. The only reference in this section is to a paper that doesn't seem to incorporate age structure at all, which is puzzling. And this is just a single example of a shortcoming in reporting their methods. I'm incredibly sorry, but this is simply subpar regarding the standard in the field of how epidemiological models are reported. The manuscript still seems, in total, to be very focused on the experience as advisors for the government and reflections on what they learned regarding the consolidation of their respective models. They do not comment on how their procedure increased forecast accuracy as compared to previous or other methods. Personally, I unfortunately cannot see the necessary novelty and cannot sufficiently judge the accuracy of the methods to justify publication as a research article in this journal. Of course this is, however, ultimately an editorial decision.

In the following some minor comments that might help in a potential revision

- spaces missing around the "and" in Eq. (B1)
- typo in first sentence of sec A.2 (usual)
- Eq. (A10) is empty
- From what the text says, Eq. (A7) carries one term $\kappa_0 S$ too much
- the authors state that their forecasts do not display a systematic bias, because the distribution of relative errors is roughly symmetrical about 0%. Yet, they also describe the systematic biases that

is also visible in Figure 1 (relative error is consistently positively biased, then negatively biased, then positively biased again). Certainly, this represents a systematic bias as their forecast either consistently overestimated or underestimated case numbers for a considerable amount of consecutive time. I would therefore suggest the authors remove this claim.

We thank the reviewer for again giving us feedback on our manuscript and helping us to further improve the draft. We are glad to see that Reviewer #2 is satisfied with our responses. In response to the remarks from reviewer #3, we have revised the manuscript to provide more technical details on the models and to more precisely state the objective of this paper. Below, we provide a point-by-point response to the remaining remarks from reviewer #3.

The authors improved the description of their methods and mostly replied to the raised comments. I applaud their efforts as advisors for the Austrian government regarding the country's response to the COVID-19 pandemic. Continuously consolidating forecasts obtained by three rather distinct methods is certainly a valuable effort to minimize critical errors in short-term science-informed consulting of legislation. Nevertheless, the initial impression that this manuscript is more suited as a commentary or perspective paper rather than an original research paper prevails.

We also want to thank the reviewer for his/her nice words and for providing constructive feedback to improve the manuscript. We actually do believe that one of the main pandemic learnings for modellers is that the modelling does not stop with producing epi-curves. Particularly when the work is supposed to support policy decisions, a sound and validated approach is necessary to communicate the model results in a way that accurately reflects strengths and limitations of the modelling approach. We strongly believe that the system that we describe in this manuscript qualifies as an original research result in the sense that we now explicitly state in the introduction:

“While the SARS-CoV-2 pandemic has triggered an explosive growth of epidemiological forecasting models, substantially less research has been performed regarding how the results of such models should be disseminated for decision support. In this work, therefore, we present the forecast and reporting system we developed based on the three independent forecasting models to support policy-making in Austria. While the individual models have been adapted from pre-existing works, our main novelty lies in developing a reporting system to communicate relevant output to non-technical experts and to inform decisions regarding strengthening or easing NPIs.”

Each of the models seems very involved and it simply does not suffice to write four pages of rough descriptions of the respective models' workings in the Supplementary Information, without any sensitivity analyses, schematic explanations, data definitions.

As stated above, the purpose of this paper is not to introduce new models – it is quite the opposite: this paper is about taking the output of models (which have mostly been described already in the literature) and describe what to do with the results of these models to better inform policy decisions. We chose the approach to differentially report model modifications that were not already described somewhere else in the literature. We apologize if we did not manage to strike this balance in a satisfactory way and have at multiple instances either clarified which parts are already described in other works and referenced this accordingly, or provided more detailed technical explanations in the SI.

As an example for such shortcomings, in the SI Sec A.2 the authors briefly mention that they extended their original model to incorporate "age structure" in "the usual (sic) way", yet they do not cite any method for what "usual way" means. I suspect I know what they mean, but they do not explicitly mention the equations that they integrated, they do not explicitly provide the contact data they used, what age stratas they decided to use. They also mention that "the parameter alpha becomes a matrix", yet falsely attribute the term "likelihood" to a rate and write that the term α_{ij} is the likelihood that a susceptible of age group j will be infected by an infected from age group i . This is not a sufficient description that enables the reader to judge the validity of the model structure, as the matrix α_{ij} has to meet certain normalization conditions and there's no way for me to check that it does. Were the authors to use canonical contact structure data as provided by the POLYMOD or COVIMOD studies, or would have cited sources that explained how they built age structure into their model (for example with the R package `socialmixr`), this might be a different story, yet they simply state that they use "mobile phone data", I assume (they don't state it) implicitly setting a number of phone calls between age groups to be proportional to the number of epidemiologically relevant face-to-face contacts between age groups and deliver no justification for this procedure, which is, however, needed, because phone calls are not face-to-face contacts. They also do not explain the mobile phone data which might not be representative of the population structure. All of this might potentially lead to very skewed contact distributions and therefore skewed results, but there is simply no way of telling, even after the revision. On top of all that, the authors never mention why exactly they incorporate age structure in the first place and how it improves their method as compared to the simple model. The only reference in this section is to a paper that doesn't seem to incorporate age structure at all, which is puzzling. And this is just a single example of a shortcoming in reporting their methods. I'm incredibly sorry, but this is simply subpar regarding the standard in the field of how epidemiological models are reported.

We have substantially expanded the SI to give a more detailed description of the implementation of age mixing in one of the models. We also provide there a discussion of why we opted to derive age mixing in the way we did (note that these decisions were made in March 2020):

"We acknowledge the limitation that phone calls are only a crude proxy for the measurement of face-to-face contacts but work with the assumption that two persons are much more likely to have close physical contact with each other if they regularly call each other compared to two individuals that never exchange phone calls. Further, this procedure allows us to measure social mixing near real-time (typically, the mobility indicators and the age matrix α_{ij} were available with a lag of two to three days), as particularly in the early phases of the pandemic it was not clear how representative social mixing data obtained via surveys from before the pandemic was for the current mixing behaviour in the society varying NPI regimes."

How the indicators from telecommunications data have been derived and how well they allow one to track responses to changes in NPIs is the subject of two separate publications [1,2] that we cite to make clear that a thorough discussion of these issues is beyond the scope of the current paper.

We have to admit that we seldom utilized the age structure of the model during the forecasts reported in this work. We implemented this age structure (again, in the early phases of the pandemic) to be ready should the need arise to answer questions that require an age structure in the model. This repeatedly was the case during our work as consortium when we were asked to provide some ad hoc evaluations of potential policy changes. However, a description of these activities is beyond the scope of this article. Note that we explicitly discuss in a paragraph in the discussion that our models indeed are much more complex than what would be needed just for the short-term forecasts, but that this structure became useful when we had to provide additional analyses.

The manuscript still seems, in total, to be very focused on the experience as advisors for the government and reflections on what they learned regarding the consolidation of their respective models. They do not comment on how their procedure increased forecast accuracy as compared to previous or other methods. Personally, I unfortunately cannot see the necessary novelty and cannot sufficiently judge the accuracy of the methods to justify publication as a research article in this journal. Of course this is, however, ultimately an editorial decision.

The main purpose of this paper is indeed a thorough description of how a monitoring and reporting system tailored toward decision support can be derived from different models and to discuss strengths and limitations that we identified in the development of such a system.

In the following some minor comments that might help in a potential revision

Thank you for your thorough reading of the text!

- spaces missing around the "and" in Eq. (B1)

Fixed.

- typo in first sentence of sec A.2 (ususal)

The section was rewritten.

- Eq. (A10) is empty

Fixed.

- From what the text says, Eq. (A7) carries one term $\kappa_0 S$ too much

Thank you for spotting this typo.

- the authors state that their forecasts do not display a systematic bias, because the distribution of relative errors is roughly symmetrical about 0%. Yet, they also describe the systematic biases that is also visible in Figure 1 (relative error is consistently positively biased, then negatively biased, then positively biased again). Certainly, this represents a systematic bias as their forecast either consistently overestimated or underestimated case numbers for a considerable amount of consecutive time. I would therefore suggest the authors remove this claim.

We reformulated this to make it clear that this claim holds only when one considers the results over the full observation window:

Nevertheless, the violin representation in Figure 1, which shows the relative differences between forecasts and reported data of all forecasts ever made, indicates that no persistent systematic over- or underestimation occurs while it cannot be ruled out that such systematic aberrations might have occurred over smaller periods of time.

References

- [1] <https://ieeexplore.ieee.org/abstract/document/9378374>
- [2] <https://www.nature.com/articles/s41598-021-97394-1>

Reviewer 4 assessed authors response to the previous reviewer comments and did not request further changes.